# Does the Olympic Agenda 2020 Have the Power to Create a New Olympic Heritage? An Analysis for the 2026 Winter Olympic Games Bid

**Martin Schnitzer ***  **and Lukas Haizinger**

Department of Sport Science, University of Innsbruck, Fürstenweg 185, A-6020 Innsbruck, Austria;
lukas.haizinger@gmx.at

* Correspondence: martin.schnitzer@uibk.ac.at; Tel.: +43-512-45865

**Abstract:** The International Olympic Committee (IOC) lacks candidates willing to host the Olympic Games (OG) and has reacted to this situation by introducing the Olympic Agenda 2020 (OA)—a reform process making the OG more attractive for potential hosts. This study analyzes whether the OA plays a crucial role for the future of the OG. We, therefore, examined the official IOC documents and feasibility studies of the 2026 Winter Olympic Games (WOG) bidders and conducted qualitative interviews with experts in the field (*n* = 15). The results reveal that the 2026 WOG hosts plan to reduce the budgets for the organization and the infrastructure costs in the host regions. As a consequence, the number and nature of the sites and venues as well as the distances between them will increase. This means that the future Olympic heritage (OH) may lay less in iconic buildings but rather focus on the attempt to fulfil the city's long-term strategies. Our analyses extend the literature by: (1) analysing the OA in view of future OG, (2) comparing experiences from past OG with those of current bidders, (3) integrating expert knowledge thanks to qualitative interviews and, finally, (4) considering new heritage concepts.

**Keywords:** Olympic Games; (modern) heritage; legacy; Olympic Agenda 2020; New Norm; IOC; mega event; event bidding

---

## 1. Introduction

Studying impacts of mega sports events has been popular in the scientific community for a few decades. However, researchers' attention has recently shifted and the interest in understanding the role of local residents in sports events as well as understanding the driving forces influencing the support for such events has increased. One reason for these new research fields is the fact that in recent years almost every public referendum on hosting the Olympic Games (OG) has failed [1,2]. Thus, understanding this opposition to the OG and the attitude towards such events as well as analysing the possible heritage they leave for hosting sites is more important than ever. Citizens have lost trust in the International Olympic Committee (IOC) but also in other international sports federations, which can be explained by corruption scandals, a lack of transparency, questionable decisions in regard to event site selection or cost overruns [3] in organizing this type of event. Furthermore, mega events may create "white elephants"—infrastructure that is not used after the event—and consequently leads to mistrust in international sports federations [4].

One of the initiatives the IOC—the owner of the OG—has implemented is the Olympic Agenda 2020 (OA) [5], which can be seen as the strategic roadmap for the future of the Olympic Movement. A key area addressed by the OA is changing the candidature procedure for the OG. Furthermore, the IOC aims to reduce the cost of bidding for and staging the OG by introducing the New Norm [6] and to transform the OG into a project that is more attractive for the host regions than it currently seems to be.

Based on the IOC's OA, which was agreed in December 2014, the aims of the study can be divided into four major research questions:

- 1: Which of the 40 recommendations of the OA for potential OG bidding and organizing committees are the most relevant?
- 2: How did (national and international) bidders for the 2026 Winter Olympic Games (WOG) interpret the OA and what are the differences compared to former host cities?
- 3: How do experts assess the OA and its potential impacts on the future development of the WOG?
- 4: What potential impact does the OA have on OG delivery and on the Olympic heritage (OH)?

To answer these research questions, the case of the 2026 WOG was selected. The article is structured as follows: The literature review encompasses the development of the WOG, the purpose of the OA as well as a theoretical introduction to the concepts of modern heritage and OH. The materials and methods used in this study are described in the subsequent section and followed by the findings of the study. Finally, the discussion and managerial implications as well as the limitations of the study and recommendations for future research are presented.

## 2. Literature Review

### 2.1. The Development and Issues of the Winter Olympic Games

In the last few decades, the content, size and complexity of the OG has dramatically increased [7,8]. Several scholars have discussed the larger budget required for staging the OG [3,9–11] while the infrastructure budget has likewise become a big issue. Thus, the constant growth of the WOG has become a real challenge [12] and nowadays the management of Olympic legacies [13,14] is one of the main characteristics of WOG. This was confirmed by the recently conducted, extensive literature review of Gaudette, Roult and Lefebvre [15]. In terms of research foci, the literature review revealed that urban impacts as well as tourism-related impacts are the most dominant topics, discussing the WOG as a catalyst for urban changes and related issues if these changes are part of a long-term strategy.

The growth of the WOG [12] undoubtedly reached a new dimension with the staging of the Sochi 2014 WOG; considering only the costs (roughly US$ 60 billion) relating to the event [16], it has become a shot across the bow against the IOC but also against potential bidders for future WOG. The negative referenda on hosting the WOG in Munich (2022), Cracow (2022), the canton of Graubünden (2026), Sion (2026), Innsbruck (2026) and Calgary (2026) but also the withdrawals of the bids from Oslo (2022), Graz (2026) and Sapporo (2026) are warning signals and can be ignored neither by the IOC nor the international sports community. The fact that, for the first time ever in its history, the IOC decided to award the Summer Olympic Games to two bidding cities (Paris (2024) and Los Angeles (2028)) reflects two major issues: firstly, justifying fruitless, cost-intensive bidding attempts has become difficult for local policy makers and secondly, the IOC is struggling to explain to its stakeholders the awarding of the OG primarily to non-European countries. Könecke and De Nooij [2] analysed the Olympic bids for the WOG 2022 and the OG 2024/28 and found that in both cases only two out of ten (WOG 2022) and seven (OG 2024/28) candidates respectively remained for the final decision in the IOC General Assembly. This trend was also a warning sign to the IOC as in previous years the interest in staging the OG had been higher.

Apart from the growth of the WOG [12,16], the complexity of organizational matters and the lack of planning certainty are central in terms of cost. On average, the cost of the OG amounts to 156% of the initially planned budget; this cost overrun has to be covered by the host city [3]. Scholars revealed that the operating budgets for the OG do not suffer from cost overruns [17], whereas the so-called non-OGOC budget—the non-event related budget including for example infrastructure—often leaves "white elephants" and is in the focus of the public debate. Thus, cost overruns for the OG are considered a high risk. Additionally, the justification of public investments and the legitimation of political decisions related to these events are based on (economic) impact studies commissioned by governments. As the majority of those studies were conducted prior to the event, they tend to have

rather optimistic assumptions (e.g., number of expected spectators, consumer spending, jobs created) and thus their results may be considered as questionable. Furthermore, these economic impact analyses have often been misapplied [18–20].

In contrast, many of the WOG's benefits, such as creating memories, know-how, networks and images [21,22], are non-pecuniary and hardly measurable. Other effects—for example the prospected growth in sports participation triggered by the OG—are yet to be proven in the scholarly debate [23,24]; thus, advocates for staging the OG are often faced with a lack of proven, fundamental arguments.

Finally, a concomitant to the aforementioned arguments can be found in the residents' support for staging the OG. The local population plays an important role in major sports events [25], not only in the context of hospitable behaviour attracting tourists. The positive attitude of residents towards the OG [26,27] may also increase the chances of being awarded the event [28]; lastly, running an Olympic bid without the support of the local population has become unimaginable in democratic countries [29,30]. However, the lack of support has manifold reasons as discussed by Könecke, Schubert and Preuß [4]. Mistrust toward the event owner but also toward the local organizing committee as well as corruption are mentioned as major concerns contributing to the negative image of the OG [31,32]. Moreover, using taxpayers' money for the organization of the OG significantly influences the outcome of an Olympic referendum [33].

## 2.2. The Olympic Agenda 2020 and the New Norm

All the facts mentioned in Section 2.1 became mosaic pieces enhancing the opposition to the IOC. As a response, Thomas Bach—after being elected IOC President—made the OA his first major initiative. It started with a public debate with over 40,000 thoughts and 1200 ideas, which were discussed in 14 working groups with experts in the field. The proposed ideas were summarized and presented in 40 recommendations for the IOC's future [5]. During the 127th IOC Session in December 2014, the Agenda was approved to become a new compass for a sustainable OG [5,34]. As the decision about hosting the 2022 WOG was taken in June 2015, the Olympic Agenda 2020 became relevant for the next bidding cycle—the 2026 WOG Bid. Six of the 40 recommendations formulated within the OA concretely focus on the planning and delivery of the OG. The IOC's [6] recommendations are:

- Shape the bidding process as an invitation (Rec. 1)
- Evaluate bid cities by assessing key opportunities and risks (Rec. 2)
- Reduce the cost of bidding (Rec. 3)
- Include sustainability in all aspects of the Olympic Games (Rec. 4)
- Reduce the cost and reinforce the flexibility of Olympic Games management (Rec. 12)
- Maximize synergies with Olympic Movement stakeholders (Rec. 13 (p. 3).

Based on the selected recommendations, the IOC established three major initiatives: firstly, a redesign of the candidature process; secondly, a new legacy approach; and thirdly, a toolkit called "7-year journey together," focusing on the planning process of the OG [6]. These initiatives were presented as the New Norm and approved at the 132nd IOC Session in February 2018. Thus, the candidature process for the OG bid should become more efficient and cost-effective for potential bidders, whilst the operational requirements for host cities delivering the Games should be reduced, the Olympic legacy should be secured throughout the event lifecycle and the costs for delivering the Games should be lowered significantly. Furthermore, the selection of the venues for staging the Olympic program should become more flexible (e.g., Olympic Village, International Broadcast Centre, Main Press Centre) and the number of training and competition venues should be reduced. Finally, the downscaling and simplification of different services should ensure cost reduction, avoiding unnecessary burdens on the local organizers [6] and thus enhancing a new OH for the hosting region.

## 2.3. (Modern) Heritage and the Olympics

In the context of mega sports events and specifically in association with the OG, the word *heritage*, deriving from the French word *héritage*, is described as equivalent to the term *legacies*; however,

MacAloon [35] argued that the word *heritage* "is more encompassing and more weighted in more contexts toward the accumulated capital of the past arriving in the present" [35] (p. 2067), whereas the term *legacies* rather defines the present's contribution to the future. Despite this discussion about legacies and heritage in the context of sports events, the word *heritage* is also used in other fields, such as urban planning and architecture.

### 2.3.1. (Modern) Heritage

Valued objects and qualities, such as historical buildings and cultural traditions that have been passed down from previous generations, are considered as heritage [36] and may be summarized with the words "precious things we want to keep." The United Nations Educational, Scientific and Cultural Organization (UNESCO) divides cultural heritage into tangible cultural heritage—movable (e.g., paintings), immovable (e.g., monuments) and underwater (e.g., shipwrecks)—and intangible cultural heritage (e.g., performing arts) as well as natural heritage such as natural sites with biological or geological formations [37]. While heritage is quite broad in its definition and also includes natural and ancient sites, modern heritage comprises "the architecture, town planning and landscape design of the 19th and 20th centuries" [38]. An example of modern heritage is the Sydney Opera House (Australia), being an important cultural centre and also landmark for the city of Sydney. Other examples could be the Grimeton Radio Station (Sweden), facilitating wireless transatlantic communication since the twenties of the last century or the city of Brasilia (Brazil), being the most popular case of a city designed on the drawing board [37].

In the context of infrastructure that has been left from hosting the WOG, Chappelet [39] names the Olympic Park in Munich (1972 WOG) and the Bergisel Ski Jump in Innsbruck (1964 and 1976 WOG) as examples. The Olympic Park Munich has become a landmark as a leisure and entertainment centre offering different activities for citizens. Even more visible, the Innsbruck Ski Jump (Figure 1) has become a major tourist attraction, especially after its renovation and architectural shaping by Zaha Hadid. Another example of such WOG infrastructure is the Olympic Ice Stadium (Figure 1) of Cortina d'Ampezzo (1956 WOG), which is still in use and is once more being considered in the Olympic bid plans of the conjoint Milan/Cortina 2026 WOG Bid. With the abovementioned definition in mind, these infrastructures can be considered as modern heritage due to their significance for the cities. Moreover, they are strongly linked with the OG. However, several issues arise when it comes to defining modern heritage, especially in terms of the assessment of the significance and the lack of distance in time. Another potential issue when discussing heritage from a monumental point of view is the fact that modern heritage gives importance to cultural processes, too [40]. Such wider concepts have been discussed in the literature [41], where the concept of modern heritage stresses the fact that it embraces "a broader range of sites, including those with associative and context value, that form the systems and networks that are traces and experiences of the processes of modernization, modernity and modernism" [41] (p. 3).

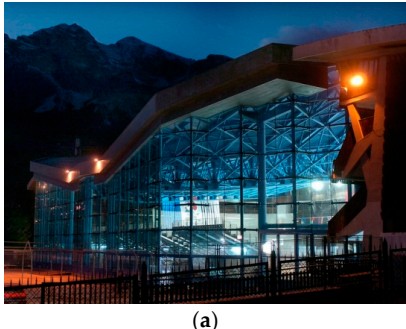　　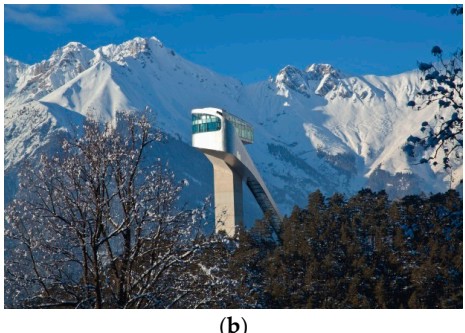

(**a**)　　　　　　　　　　　　　　　　　　　　　　　(**b**)

**Figure 1.** (**a**) The Olympic Ice Stadium (1956) in Cortina d'Ampezzo (ITA) and (**b**) the Bergisel Ski Jump of the 1964 and 1976 WOG in Innsbruck (AUT), renovated in 2001 (Photo Credits: (**a**) Stefano Zardini; (**b**) Innsbruck Tourismus, Christof Lackner).

2.3.2. (Heritage) Sports Events

As shown in Section 2.3.1, the word *heritage* may be used in the context of sports events. However, Chappelet [42] mentioned existing heritage sports events (HSE)—which can be defined as events generally involving a single sport—that have been held in the same place for many years and have been hosted regularly since their foundation—usually each year, sometimes biannually, without interruptions other than under exceptional circumstances. Furthermore, those events have to be recognized as institutions [43]. If we consider the notion of heritage events, the concept of a "place of memory" [44] could be attributed to the OG, referring to the OG's tradition as a one-off event—and as a heritage event—, because it strongly links the event to the place in which it is held. An example for this may be the following quote by the former Governor of Utah, Michael O. Leavitt [45]:

> *About 10 years ago, I was representing the state in Europe and I flew in for a plane change in a small Austrian village called Innsbruck. As I landed, I thought, "What is it about Innsbruck I know? Oh, it was an Olympic city." Innsbruck is a village. I was 13 years old when this happened. It is branded in my mind as a quality place and as an Olympic city. Once you are an Olympic city, you are never the same.*

Even though the OG may become a heritage event due to their huge impact, literature has mainly discussed HSE as recurring events building the expertise needed to ensure the event's continuing success over the years and thus becoming a brand, such as the Wimbledon Championships, which were already established in 1877 or the Vasaloppet in Sweden, which has been taking place since 1922 [46]. Based on the study by Pinson [47], the characteristics of HSE lie in their persistence and sustainability in terms of the location in which they take place, in their recurrence, in local governance and thus in their longevity as constituent elements. Furthermore, their unique character (distinguishing them from other events), specific hard features (e.g., infrastructure), soft features (e.g., know-how) and the disposition of recipients, make such sports events authentic and thus they may become HSEs. Consequently, HSEs turn into territorial resources that strengthen the local economy, attract tourists and may also have a positive impact on the quality of life and the territory's overall appeal. The narrative of a sports event can be considered as a necessary condition for identifying the event as an HSE [47].

Just recently, literature has again discussed [48] the meaning of small-scale HSEs as a sustainable development tool for tourism and showed that small-scale HSEs have huge potential to create social legacies and thus become part of the cultural heritage of a given region. Whether the same concept also applies to the WOG, lending the Games a new OH, will be discussed in this article.

2.3.3. Olympic Heritage

One of the compulsory characteristics of HSEs is their sustainability; Olympic Heritage, by definition, needs to be considered as something long-lasting with positive aspects in its nature, constituting a distinct, already noted long-term legacy. Over time, various types of legacies have been identified in the literature. Legacies may have economic, physical, socio-cultural, psychological and political facets [49,50]. Scholars mainly agree on the definition of legacies as "all the planned and unplanned, positive and negative, intangible and tangible structures created by and in connection with a sports event that remain for a longer period than the event itself" [51] (p. 211) and that they may also depend on the attitude of each of its various stakeholders. Hence, we could argue that an event-related heritage originates when long-term positive event legacies occur. Chappelet [39] defines hard and soft heritage as being personal (linked to an individual associated with the event) or territorial (specific to the host city or territory; therefore the event cannot be replicated elsewhere). Thus, four categories of heritage may be distinguished: the territorial hard heritage (e.g., Olympic sports facilities), the personal hard heritage (e.g., a volunteer's uniform), the territorial soft heritage (e.g., a new destination image) and the personal soft heritage (e.g., knowledge gained by an employee involved in the event's organization). This shows that—irrespective of Olympic venues as tangible

(hard) heritage—the intangible (soft) heritage should also be seen as an essential part of the heritage concept in the Olympic context and needs to be nurtured [39].

MacAloon [35] discussed the misuse of and/or confusion about the term *legacy*; however, he concluded that the term *Olympic heritage* should be understood as the total sum of accumulated Olympic cultural, historical, political, moral and symbolic capital. Depending on their positive or negative nature, legacies may increase or decrease this capital. By integrating the concept of modern heritage, the modern OG—being just older than 120 years—may serve as an example of (modern) OH; not only due to their iconic monuments, such as Beijing's Birds' Nest, the Montjuïc Communications Tower designed by Santiago Calatrava or the cost-intensive Olympic Stadium in Montreal but also because of urban planning projects realized in East London or touristic transformation processes such as Barcelona, becoming one of Europe's top hubs for MICE tourism [52].

Irrespective of the OH and cases showing the benefits the OG may have, positive legacies of the OG are often presented in an exaggerated manner, whereas any negative impacts are understated [53]. Hence, negative effects, such as missing sustainability [54], cost overruns [3,10], the OG's increasing complexity [55,56] and commercialization as well as missing transparency [4] may find fertile ground, as presented in Section 2.1.

Consequently, considering the discussion about the OH and the severe issues the Olympic Movement is currently facing, the question arises whether the OA could be the saviour of the IOC, heralding a new era in Olympic history and creating a new, modern OH?

## 3. Materials and Methods

In order to give answers to the research questions and to discuss the different approaches of the literature in this field, a document analysis and guided qualitative interviews were undertaken.

### 3.1. Document Analysis

For conducting the extensive document analysis we gained access to several types of relevant documents via the Olympic Studies Centre in Lausanne, the official website of the IOC (www.olympic.org), the websites of (former) candidates and other sources (e.g., online access to the websites of national and international newspapers). Furthermore, in April 2018, we directly approached the National Olympic Committees of the following countries (in alphabetic order): Austria, Canada, Japan, Sweden, Switzerland and Turkey; they had nominated an international candidate. Finally, we worked with the following documents:

- Official IOC documents: Evaluation Commission Report for the 2026 WOG; IOC Olympic Agenda 2020; Olympic Games: the New Norm; Report of the 2026 WOG IOC Working Group;
- Final reports of previous WOG: Salt Lake City 2002, Turin 2006, Vancouver 2010, Sochi 2014 and PyeongChang 2018;
- Feasibility studies for hosting the 2026 WOG (in alphabetic order): Calgary 2026, Cortina d'Ampezzo 2026, Graubünden 2026, Graz 2026, Innsbruck 2026, Sion 2026, Stockholm 2026 and Turin 2026.

Based on the method of Atteslander [57], the accurate and intensive reading of the documents constituted the first step of the document analysis. In a second step, the documents were screened once more, considering their relevance to the research questions. In order to ensure the explanatory power, the official final reports of former WOG host cities were studied in addition to the feasibility studies for the 2026 WOG. The 40 recommendations of the IOC's OA became the most relevant anchor points for building different categories of interest.

### 3.2. Qualitative Analysis

The qualitative interviews discussed the OA and its possible impacts from different stakeholders' views and should give insights into the potential this endeavour offers for actual hosts and the

development of heritage in these regions. The present study's intention is to show the focal points of the OA on the one hand and the impacts it may have on future OG and on potential 2026 WOG host cities on the other hand.

We conducted a convenient and purposeful sampling procedure [58] as we wanted to interview key stakeholders involved in the bidding and staging of potential WOG. The face-to-face, semi-structured interviews (see Table 1) took place in person, via telephone or Skype meetings. In four cases, the interviewee preferred to answer the questions from the interview guide in writing.

**Table 1.** Information on the interviews with the experts.

| Function/Role of the Interview Partner | Type of Institution (Country) | Date & Interview Duration (in Min) | Method (Author Responsible for Interview) | Acronym |
|---|---|---|---|---|
| Consultant | For-profit event company (AUT, GER) | 11 December 2017 (96 min) | in person (author 2) | I1 |
| Managing director | Public event company (AUT) | 11 December 2017 (35 min) | in person (author2) | I2 |
| Project manager | National Olympic Committee (AUT) | 12 December 2017 - | in writing (author 2) | I3 |
| Scholar, consultant | Private university (AUT) | 15 December 2017 (10 min) | by telephone (author 2) | I4 |
| Scholar, consultant | Public university (SWI) | 3 January 2018 - | in writing (author 2) | I5 |
| Division manager, event manager | National winter sports federation (AUT) | 4 January 2018 (37 min) | by telephone (author 2) | I6 |
| Mayor | Bidding city authority (AUT) | 12 January 2018 (20 min) | in person (author 2) | I7 |
| Managing director, president | Olympic venue, national winter sports federation (AUT) | 22 January 2018 (26 min) | by telephone (author 2) | I8 |
| Managing director, consultant | For-profit event company (GER) | 24 August 2018 (25 min) | by telephone (author 2) | I9 |
| Consultant | For-profit event company (FIN) | 3 September 2018 (35 min) | via Skype (author 2) | I10 |
| Secretary general | Multi-sport event, international single-sport event (ITA) | 18 September 2018 (35 min) | via Skype (author 2) | I11 |
| Secretary general | International winter sports federation (UK) | 18 September 2018 - | in writing (author 1) | I12 |
| IOC member | International multi-sport federation (CAN) | 3 October 2018 (64 min) | via Skype (author 1) | I13 |
| Scholar | Public university (CAN) | 25 October 2018 (30 min) | via Skype (author 1) | I14 |
| Secretary general | International single-sport event (ITA) | 27 October 2018 - | in writing (author 1) | I15 |

The interviews (*N* = 15) were conducted in the English or German language—depending on the nationality of the interviewee—between 12 December 2017 and 31 October 2018. In total, 13 interviews were required to reach theoretical saturation as prescribed by Guest, Bunce and Johnson [59]; however, we conducted 15 interviews to provide both theoretical saturation and a satisfying variety of international experts. The interviews lasted between 10 and 96 min. All interviews were developed and conducted according to the ethical guidelines and criteria stated by Patton [60]. As Table 1 describes, all interviewed persons can be seen as experts as they have practical and/or theoretical knowledge about the OA, the development of the OG and the potential future development of the OG. The interviews ended with the open process-feedback question: "Do you have anything else to add?" Afterwards, the interviews were transcribed verbatim in their original language by one

of the bilingual authors. The quotes were translated solely for the purpose of reporting the findings to preserve the integrity of the data.

The raw material was read and encoded, following the qualitative content analysis of Fenzl and Mayring [61]. Quotes, themes and paraphrased quotes were identified individually when they seemed to be a considerable point or thought. The emerging findings were compared with the data once more to verify correct comprehension and were also discussed with colleagues. The interviews focused on the following aspects:

- The development of the OG;
- The current situation and issues facing the IOC regarding the OG;
- The OA (aims and relevance of the recommendations with regard to the OG);
- The future development of the OG and the role of the OA.

## 4. Results

### 4.1. Document Analysis Findings

The careful analysis of the OA [5] revealed six major topics into which the recommendations can be divided, namely:

- Cost effectiveness (recommendations **3**, **12**, **13**);
- Sustainability (recommendations **1**, **2**, **4**, 5);
- Transparency/Media (recommendations 19, 20, 23, 28, 29, 30, 31, 39);
- Sports (recommendations 6, 7, 8, 9, 10, 11, 15, 16, 17, 18, 25, 34);
- Networking (recommendations 26, 27 33, 35);
- Others (recommendations 14, 21, 22, 24, 32, 36, 38, 40).

With regard to the organization of the OG, the New Norm [6] focused on the six recommendations highlighted above in bold font and briefly presented in Section 2.2; these recommendations are attributed to aspects of cost effectiveness in terms of bidding for and staging the WOG and to aspects ensuring the highest possible sustainability of the event. In order to operationalize these recommendations and the three major initiatives [6], Table 2 summarizes them in three major topics: One topic relates to reducing the cost of bidding for the WOG, the second to the delivery of the WOG—thus considering the local organizing committees' budget (OCOG budget) and the investment budgets for venues, transport infrastructure and similar, covered by taxpayers (non-OCOG budget)—and the third is dedicated to measures ensuring the implementation of the Legacy Strategic Approach [6], which focuses on reducing, replacing or temporarily using competition and training venues, Olympic villages and/or other facilities, such as the International Broadcast Centre and Main Press Centre. Table 2 compares the (bid) budgets, the different kind of venues, their nature (existing, newly built and temporarily built) and (Olympic) sites such as venues outside the host city as well as the respective distances (maximum and average) between these sites. The comparison not only includes potential WOG bidders but also compares them to former and future Olympic hosts (2002–2022) in order to analyse differences.

**Table 2.** Information about former and future WOG host and bidding cities with specific focus on their budgets and Olympic venues.

| (Potential/Actual) Host City Staging the Winter Olympic Games | | | Budget in USD Billions (Calculated on the Basis of the Year 2018) | | | Venues (Number) and Distances to the Host City (in km) | | | | | | | |
|---|---|---|---|---|---|---|---|---|---|---|---|---|---|
| City & Year | Country | Host/Bid | Bid Budget [1] | OGOC Budget [1] | Non-OGOC Budget [1] | Sites | Olympic Villages | Competition Venues | Existing Venues | New Venues | Temporary Venues | Max. Distance | Avg. Distance |
| **Salt Lake City 2002** | **USA** | Host city | N/A | 2.722 | N/A | 6 | 1 | 10 | 0 | 5 | 5 | 113 | 49 |
| **Turin 2006** | **ITA** | Host city | N/A | 4.693 | 5.180 | 7 | 3 | 13 | 2 | 8 | 5 | 78 | 33 |
| **Vancouver 2010** | **CAN** | Host city | 0.018 | 2.032 | 0.671 | 4 | 2 | 9 | 4 | 5 | 0 | 107 | 55 |
| **Sochi 2014** | **RUS** | Host city | 0.016 | 4.530 | 54.030 | 2 | 2 | 10 | 0 | 10 | 0 | 40 | 40 |
| **PyeongChang 2018** | **KOR** | Host city | N/A | 1.531 | 1.900 | 2 | 2 | 13 | 7 | 5 | 1 | 62 | 62 |
| **Beijing 2022** | **CHN** | Host city | N/A | 1.558 | 1.600 | 3 | 3 | 12 | 6 | 0 | 6 | 163 | 112 |
| *Average 2002–2022* | | | *0.017* | *2.338* | *12.676* | *4* | *2* | *11* | *3* | *6* | *3* | *94* | *59* |
| **Graubünden 2026** | **SWI** | National bid (W) | 0.025 | 1.669 | N/A | 9 | 4 | 17 | 14 | 3 | 0 | 143 | 70 |
| **Innsbruck 2026** | **AUT** | National bid (W) | 0.017 | 1.175 | N/A | 12 | 1 + 5 | 13 | 10 | 0 | 3 | 518 | 181 |
| **Turin 2026** | **ITA** | National bid (W) | 0.007 | 1.411 | 1.148 | 6 | 3 | 24 | 24 | 1 | 0 | 78 | 31 |
| **Cortina 2026** | **ITA** | National bid (M) | 0.007 | 1.451 | 0.449 | 7 | 3 | 11 | 9 | 2 | 0 | 103 | 57 |
| **Sion 2026** | **SWI** | Official bidder (W) | 0.025 | 1.761 | 0.412 | 15 | 8 | 19 | 6 | 7 | 6 | 245 | 80 |
| **Graz 2026** | **AUT** | Official bidder (W) | 0.010 | 1.359 | N/A | 10 | 1 + 5 | 13 | 12 | 0 | 1 | 292 | 122 |
| **Erzurum 2026** | **TUR** | Official bidder (N) | N/A | N/A | N/A | 4 | 3 | 10 | 8 | 2 | 0 | 680 | 136 |
| **Milan/Cortina 2026** | **ITA** | Official bidder | N/A | N/A | 0.443 | 6 | 4 | 12 | 8 | 1 | 1 | 415 | 344 |
| **Calgary 2026** | **CAN** | Official bidder (W) | 0.024 | 2.417 | 2.469 | 3 | 5 | 9 | 7 | 0 | 2 | 635 | 453 |
| **Stockholm 2026** | **SWE** | Official bidder | N/A | 1.623 | N/A | 4 | 4 | 11 | 9 | 2 | N/A | 525 | 375 |
| *Average 2026* | | | *0.016* | *1.608* | *0.984* | *8* | *5* | *14* | *11* | *2* | *1* | *363* | *185* |

[1] Data on budget and venues retrieved from [10,11,16,17,53,62–82]. The exchange rate and inflation were calculated based on the systematic discussion in the literature [83]; N/A = not applicable; W = withdrawn bid; M = merged bid; N = not considered by the IOC after the 2026 WOG Working Group Report.

Table 2 shows the budgets of the recent host cities, bidders and national bids: the average bid budget was USD 17 million, the average cost of staging the WOG (measured on the basis of the OGOC's operating budget) was USD 2.844 billion and the average budget for investments (mainly for housing, venues and transport infrastructure) amounted to USD 12.676 billion (interesting note: without the Sochi 2014 WOG, the average non-OGOC budget was only USD 2.338 billion). The bid budgets for the 2026 WOG bids amounted to an average of USD 16.5 million and are thus estimated to be slightly lower than the budgets of the past bids; however, the estimated OCOG budget for the 2026 WOG hosts averages USD 1.608 billion and is hence significantly lower than the respective budgets of the 2002–2022 host cities (USD 2.844 billion on average). These differences are even more notable if we consider the 2026 WOG's non-OCOG budgets, which are expected to be below USD 1 billion (~USD 984 million on average). Interestingly, three bidders (the canton of Graubünden, Innsbruck and Graz) even argued in their feasibility studies that there would be no cost for taxpayers (non-OGOC budget) as all investments related to the WOG had been planned anyway [70,81,84]. If we consider the typical WOG cost overruns and include them in the estimated budgets, the cost of the WOG staged in the last two decades (2002–2022) was still significantly higher than that expected for the 2026 WOG (compare [9]).

Regarding the venues, a similar pattern can be found. While in the 2002–2022 WOG the average number of sites (towns outside the host city where competitions took place) was 4, the 2026 WOG Bids planned 8 sites on average. The number of Olympic villages—the accommodation sites for Olympic athletes—increased from an average of 2 to 5. A similar pattern seems to apply to the other venues (increase in competition venues: 11 to 14 on average; increase in use of existing venues: 3 to 11 on average; decrease in new venues: 6 to 2 on average; and decrease in temporary venues: 6 to 1 on average). As a result of these developments, the distances between the host cities and the venues/sites will increase significantly: the average maximum distance has risen from 94 km (for the 2002–2022 WOG) to 363 km (for the 2026 WOG Bids) and the average mean distance from 58 km (for the 2002–2022 WOG) to 185 km (for the 2026 WOG Bids).

Figure 2 shows the Venue Masterplan of the conjoint bid of Milan and Cortina d'Ampezzo. The travelling time from Milan to Cortina d'Ampezzo is roughly four to five hours by car and the distance of 400 km is much greater than in any of the previous WOG.

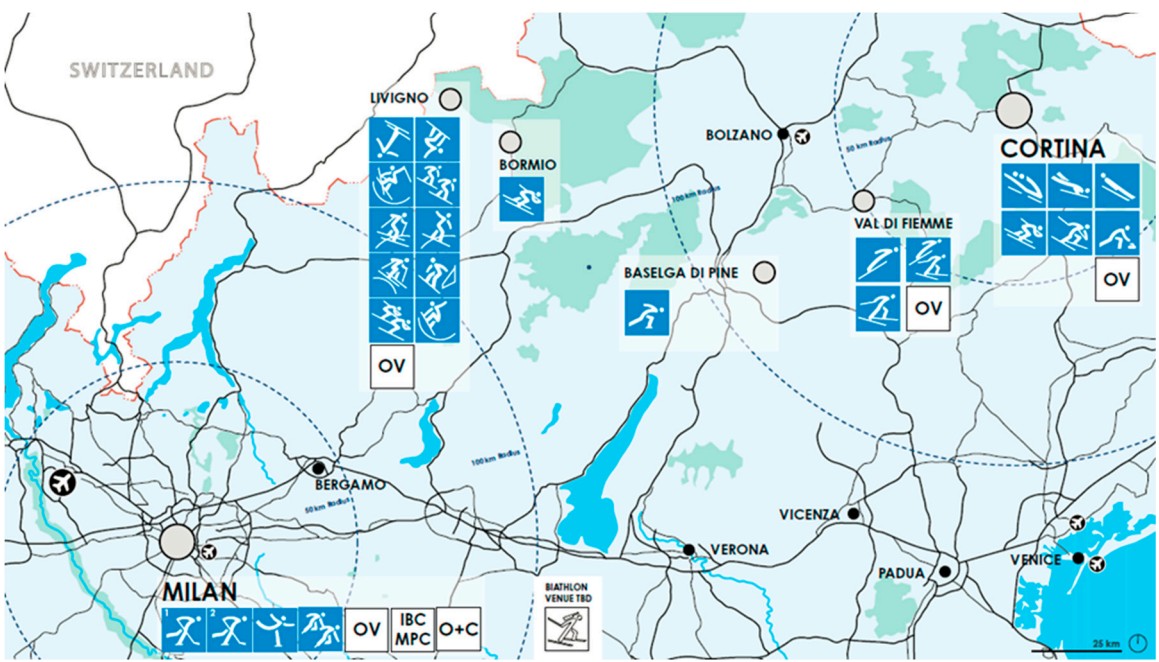

**Figure 2.** Venue Masterplan of the Milan/Cortina d'Ampezzo bid for hosting the 2026 WOG, indicating the proposed sites and venues [85] (p. 155).

*4.2. Expert Interviews—Findings*

The second part of the results section—the analysis of the expert interviews—can be split into three parts: firstly, the development of the OG and current issues the IOC is facing regarding the OG; secondly, the influence the OA may have on the OG and the question of which of the recommendations may be crucial for the OG; and thirdly, the future development of the OG and thus the emergence of potential new modern heritage related to the OG.

4.2.1. Development of the Olympic Games and Current Situation

Experts see both positive and negative developments with respect to the OG. They can be summarized in two major topics: one relates to potential benefits for the (international) sports federations and the other to a complex of negative topics, such as corruption, doping and cost overruns at past OG, as well as to the IOC's poor reputation as the following quotes indicate:

*"Since the Games in Lillehammer, the Winter Olympics have grown too quickly or been too inflexible."* **(I4)**

*"In sports there is much more money to earn and everywhere, where a lot of money is involved, a lot of trickery is done. And then there are issues, such as corruption, bribery and doping."* **(I1)**

*"The selection of the various cities has, as I say now, shifted to promising markets or to countries that can afford it a little bit easier. Both politically and financially. I see that very critically."* **(I2)**

*"First of all, there is the reduced credibility of the IOC. In addition, the gigantism of the past years. Thirdly, there simply are socio-political reasons, especially in Central Europe, where the necessity of hosting Olympic Games in the region is viewed very critically—for local political, personal and emotional reasons."* **(I7)**

*"The local community has become more critical and mature and is tired of greenwashing and 'white elephants.'"* **(I11)**

*"After the successful Games of '96 and '00, the OG has grown dramatically and thus the cities' interest in staging. This also created high costs, which nowadays are seen critically."* **(I9)**

*"The size and scope: finding the balance between cutting edge sports and traditional sports while minimizing structures and costs will be an ongoing future challenge."* **(I12)**

*"The IOC has become one of the most important financiers of the National Olympic Committees and international sports federations."* **(I13)**

Speaking of the OG's current situation and issues the IOC is facing in regard to the OG, the experts named various aspects, such as size and feasibility, lack of communication or opportunity costs, as the following quotes show:

*"The IOC has to make sure that the OG are manageable; especially in terms of size and costs. Lately, the Games have become too gigantic and cost a lot of money. On the other hand, they must keep their uniqueness and ensure that organizing the Games is worth to the host."* **(I5)**

*"I believe that 90% of the population doesn't even realize what the Olympic Games mean for a host city, also on the revenue side. You could tell that by the argument: 'You can use the money for much better things.' But the money doesn't come for anything else."* **(I1)**

*"I believe that the OG need to become easier to finance and more sustainable—in short: more human."* **(I11)**

*"Games like Sochi were negative for the overall development, especially in democratic countries. Furthermore, the lack of communication with the population has become a big issue."* **(I10)**

*"The biggest 'to-dos' for the IOC are winning back trust and communicating—What are the benefits of the OG for the population?"* **(I9)**

*"The Games got bigger and bigger and now we have a gigantism issue. At the same time, the IOC is always being reactive."* **(I14)**

*"One of the issues of the IOC is that there is a lot of fake news around the Games."* **(I13)**

### 4.2.2. Olympic Agenda 2020 from an Experts' Perspective

Experts welcome the OA initiative and believe that a turnaround may be possible; however, they criticize that the IOC has not been behaving actively but reactively and thus has been at risk of losing control. Therefore, the experts recommend: downscaling the WOG, being more flexible, fighting cost overruns and gigantism as well as avoiding "white elephants."

*"As the most important thing, I simply see the scale—that is, I think, the most appropriate term."* **(I8)**

*"One very important aspect is the recommendation to use existing facilities. That also includes the fact that the Games can be spread over larger areas or even two different countries."* **(I5)**

*"For me, the most important aspects are those that put the sports back in the focus and make the organization of the Games affordable again."* **(I3)**

*"The OA is still not concrete enough."* **(I2)**

*"The costs of bidding are much less relevant than the ones for staging; the New Norm is much more relevant than the OA itself."* **(I10)**

*"In the end, it is about less money, better legacy, better involvement of the people. If this is reached, the OA was successful."* **(I15)**

*"Why did the OA not come out when the scandals happened? Why did it take 30 years for change or a document to be published with recommendations? Why are we seeing some little movement only now? This is again the point of being reactive. The concept of the Agenda is good but there are no new ideas and it is just slow moving."* **(I14)**

*"Well, the Agenda 2020 was important for the new IOC President Thomas Bach but there isn't really something new in it."* **(I13)**

When it comes to the future of the OG, experts see the reduction of costs for bidding and staging the OG as well as becoming flexible in the Games' management as the most crucial recommendations formulated in the OA.

*"So when your credibility is so badly tarnished, you can only regain it if you are 100% transparent. And when the governance structures become more open."* **(I2)**

*"Giving host regions the possibility to develop infrastructure will also be important for the future."* **(I10)**

*"The cases of Sochi and PyeongChang are worst practice examples, which make evident, how important the Agenda 2020 is. In fact, around 80% of the venues for the 2026 WOG already exist."* **(I9)**

*"By emphasizing the use of existing facilities and infrastructure in other places in the host nation or region, either in the same or a neighbouring country, the hosts will deal with controlling infrastructure costs and potential post-Games 'white elephants.'"* **(I12)**

Finally, experts were also asked for their opinion on how long it will take for the results of the OA to become visible. Most experts see the Paris 2024 OG as the first benchmark, where at least parts of the OA should be implemented.

> *"The goals of the Agenda 2020 are not set for 2020. It is a goal for eternity. It must be updated again and again. For me, those are not medium-term goals either. It is a long process that has to be worked on again and again; one has to be open and to communicate more."* **(I1)**

> *"I think because of the European environment, because of summer, because of the concept I have seen and because of the absolute will and necessity to do something, Paris will be a benchmark."* **(I2)**

> *"Maybe we'll see some consequences already in 2020 and 2022. It will take at least four years or maybe even eight years until all recommendations will be realized because the Games are awarded seven years in advance."* **(I5)**

> *"Credibility is the only way. When we have succeeded in implementing the Agenda 2020 for the first time, credibility will certainly rise. The decisive point for the Winter Olympics will be the awarding of the 2026 and 2030 Games."* **(I7)**

### 4.2.3. Future of the Olympic Games

The last part of the expert interviews was dedicated to looking into the future and discussing what future OG might be like. The experts have different approaches and opinions, which may be grouped into two blocks. One group of experts is very sceptical about the implementation of the OG, the other group is more positive, mentioning an increase of interest in sports and sports infrastructure. However, both groups think it will take years to convince the local population of the benefits of the OG; there have been too many negative examples and thus the potential host community does not believe in the good of the OG.

> *"I would say that all major sports should be part of the Olympics and some obsolete ones could be dropped for the sake of reducing the gigantism and costs."* **(I5)**

> *"So definitely a downsizing. Also that the spectator capacities of the venues are made more flexible, the number of sports is reduced and old-school sports attracting fewer spectators are dropped from the program."* **(I3)**

> *"I think it is simply necessary to reduce the standards a little bit; otherwise the infrastructure madness will continue and that makes it very difficult to prove sustainability."* **(I7)**

> *"Future OG need to be more creative and tested like the IOC does with hosts of the Youth OG."* **(I11)**

> *"I think the Agenda 2020 is a really honest attempt of the IOC to develop the OG but it may take some time; however, serious No-Olympia campaigns need to be taken into consideration by the IOC and the potential hosts."* **(I10)**

> *"The OG of tomorrow will come back to the core markets; will have individual approaches serving the needs of the hosting regions and the needs of the athletes."* **(I9)**

> *"They will continue to be THE sporting event that sets the tone for the rest of the sports world to follow. Striking a balance between tradition and innovation, whilst upholding the highest values of integrity."* **(I12)**

> *"I would like to see the OG and the IOC at the forefront of using every new developing tool that has a positive impact and not just looking at the economics of it—looking at the social economic and environmental impacts."* **(I14)**

> *"In principle, the Games should become friendlier and more emotional again, as we have experienced them for example in Lillehammer."* **(I6)**

## 5. Discussion

Based on the findings, which are summarized in Table 3, we could identify positive and negative relevant developments of the OG and their potentials thanks to the OA. The single research questions are discussed in the following subchapters.

**Table 3.** Summary of the findings.

| | Positive Aspects | Negative Aspects |
|---|---|---|
| **Development and current situation of the Olympic Games** | High demand, TV and marketing rights; Transfer of income to sports federations; Structural and financial development; Attractive Olympic brand; IOC is the benchmark for the sports world | Sustainability and legacy as buzzwords; Questionable selection process, too much political influence; Fast growth, high costs; Games are too complex; Corruption, doping, gigantism, lack of transparency; Decrease in candidates, negative referenda; Bad publicity, poor reputation of IOC; IOC is too reactive and slow in changing processes; Lack of trust in international sports federations; Communication issues, concepts are not understood; Agenda 2020 is not concrete enough; Success of Agenda 2020 will take several years |
| **Potentials of the Olympic Games thanks to the IOC Olympic Agenda 2020:** | | |
| **The future of the Olympic Games** | Downsizing in terms of venue capacities and the size of Olympic village(s); Flexibility in the number of sites, venues and involved countries; Reduction of costs and size due to lower infrastructure budgets; More permanent venues even though their spectator capacities will be much smaller; Less standardization and higher potential for innovation; Possibility to shape host cities'/countries' legacies; Chance to focus more on sports; Alternating, new disciplines, older sports formats may disappear; | |

### 5.1. The Olympic Agenda 2020 and Its Implementation

Even though several experts criticized the OA as being insufficiently concrete, the New Norm [6] does not lack specific examples of where and how the OA—for instance in reducing the OCOG budgets—might be useful. However, after the negative referendum for the Calgary 2026 WOG Bid, the Olympic journalist Robert Livingstone summarized: "Agenda 2020 isn't working. It has become the punchline of a Twitter post. It is time for the IOC to change everything." [78] (p. 1). The results showed that costs have decreased in all areas (bidding, staging and building infrastructure); however, this has not been the case in the past, as revealed in the literature [3]. Analysing the budgets of the local organizing committees (LOC) for the FIFA World Cup, Preuß and Schnitzer [83] discussed the reasons for budget increases and argued that bidding cities often keep the LOC budgets low for strategic reasons in order to appear more attractive to the event owner. As confirmed by all experts, the implementation of the OA aims at reducing the Games' costs by involving more sites in different areas of the host regions, even in other, neighbouring countries; however, this may increase the complexity of co-hosting events [86].

Therefore, in regard to the first research question on the most relevant recommendations of the OA, reducing costs and enabling the creation of legacies tailored to the host regions are doubtlessly of the greatest importance. As mentioned in Interview I10, the cost of bidding will be less important than others; however, it could send a meaningful signal and message to taxpayers.

Regarding the second research question, dealing with areas of implementation it can be noted that the national and international 2026 WOG bidders have taken into account the possibilities offered by the OA to a maximum extent; the results are reflected in the budgets and the use of venues as shown in Table 1. The feasibility studies showed that reducing seat capacities at the competition venues or even holding the Opening Ceremony at several venues [84],thus avoiding the construction of oversized stadiums, may become innovative concepts in the long term and may give rise to new approaches to OG organization. This example but also many others, may lead to a new understanding of OH.

However, Livingstone [78] almost hopelessly argues that taxpayers are penalizing the IOC for its past mistakes with negative referenda.

> *The truth is the IOC really has changed. The bid process is completely different than it was in those days of overspending. Cities are encouraged to spend less and the IOC is willing to pitch in and help make that happen. They really, really are. But it won't matter.* [78] (p. 2)

It seems that the IOC's new philosophy has not yet reached local host communities and is thus missing an important stakeholder.

*5.2. The Future Olympic Games—New Olympic Heritage?*

Concerning the third research question on the importance of the OA for the future of the OG, the experts' statements generally tended to be positive, even though some may be interpreted in an ironic way. Future OG might be smaller in terms of investment but bigger in terms of involving larger territories. The two remaining bidders for the WOG 2026 definitely corroborate this trend. The possibilities given by the OA will be outbid. Future hosts may sit on the long branch—at least in the short run—and will not participate in the rat race [83]. Will the IOC become a "lame duck," simply fulfilling the hosts' interests or is the current situation welcome medicine, the results of which may become visible only in the next five to ten years? Experts see the Paris 2024 OG as a milestone and the first measure of the OA's effects. Even so, it may well take longer for the impact to be felt. The newly structured OG bidding procedure, in which the IOC plays a supporting role, also intends to help adapt the size and requirements of the Olympic Games to the regional settings and vice versa. Even though the experts agreed that the OA may help to brush up the image of the IOC, they also believe that this requires—first and foremost—the implementation of a new concept to achieve the OA's goals. Some recommendations have already been implemented but, as the experts indicate, this is currently appreciated only by insiders. A first tangible result of the OA as regards the reduction of bidding costs, is the pragmatic handling of the two remaining bidders for the OG 2024/28 and the double-award by the IOC. Thus, both candidates won, even though the IOC's price (in terms of TV and marketing rights contribution) was high.

When discussing the fourth research question, examining the OA's potential impact on OG delivery and OH, it is necessary to start with the current premise that the IOC has to accept the consequences for of its past excessive culture, which damaged the Olympic brand, as Livingstone [78] summarized: "If the IOC can't rebuild its credibility, the message of OA is worthless and stakeholders will form their own opinions" (p. 3). This statement is supported by the experts' opinion. On the one hand, the lack of credibility (I7) and, on the other hand, the fact that the community has become more critical (I11) have led to an increase in opposition to staging the OG [87]. Due to the lack of local support for an Olympic bid [88], the fall in the number of bidding cities has reached a new dimension.

Is the OA really a failure? No, not necessarily—provided that the 2026 WOG host succeeds in making the OA work; applying the OA could thus become a benchmark for future OG. It will, however, fail if the IOC does not have a reliable partner for the 2026 WOG. If the second case becomes a reality, the IOC will need to push the reset button. If the first case becomes a reality, the opportunities for creating a new OH will be abundant: the OG would become less expensive and more sustainable by using mainly existing infrastructure; the Games would involve a wider territory and not just a single city; and they would have the potential to spread the Olympic values and ideals to a larger community. This probably would not have an influence on the Olympic values [89] but on other intangible legacies, such as new networks within the hosting regions and cross-border collaborations [86]. Community cohesion may be even more relevant if the community does not just think about the amount of tax spent on "white elephants." The fact that the OG's financial burden would be shouldered by more partners could be important for creating a new OH. In fact, as revealed in the discussion about modern heritage, heritage may also have intangible elements. It is very likely that future Olympic host cities concentrate on taxpayer-friendly rather than on monument-driven Games. Consequently, modern OH may be more

intangible than ever; however, there is also an opportunity to create another form of tangible landmark by avoiding spending on doubtful buildings.

*5.3. Implications and Limitations*

The outcomes of this research revealed several implications that are worthy of consideration by policy makers and the wider sports community. As one of the interviewed experts (I12) underlined, the IOC is a typical "first mover": The past has shown that the IOC sets the benchmarks for all other federations. If the OA fails, not only the IOC but also other sports federations will be affected.

Furthermore, it is imperative that key decision makers accurately inform the local communities about the possible benefits but also the risks the OG may hold for them [90,91]. Another implication is that the sustainable effects of events—in this context the OG—can be achieved only if the OG are embedded in a long-term strategy [92]; this means: if the OG do not support the host regions' long-term strategy, they may become a risky endeavour. Finally, the experts (I13, I14) indicated that good communication between the IOC, the local key stakeholders and the local community is decisive for gaining the residents' support.

Finally, some limitations of this study deserve attention. First of all, it has not yet been decided which country is going to host the 2026 WOG; thus, due to certain decisions, many considerations may become obsolete (e.g., what would happen if Sweden and Italy withdrew their bids?). Secondly, there were some issues with the data: the data was not easy to collect and not always at our disposal (see N/A); moreover, some data given in the analysed documents did not consider all types of costs [93]. In order to calculate the budgets at the level of the year 2018, various methods applied in the literature [83] were considered. Nevertheless, inflation and exchange rates may vary and are probably not accurate enough. Furthermore, limitations regarding the qualitative interviews must be mentioned. Even though a stakeholder analysis was conducted and the interviewees were selected carefully, a selection bias in the choice of interviewees cannot be excluded. Lastly, many ideas that were put forward in the bidding files may not be accepted by the IOC or not considered as feasible. This means that the comparison of the figures listed in Table 1 might not hold true over the course of time and are likely to become subject to change.

*5.4. Conclusions*

To conclude this study, several points have to be highlighted: First of all, many issues relating to the OA and to the future of the OG depend on decisions about the 2026 WOG; finally, the Olympic Movement is at an important crossroads. As one of the experts (I11) mentioned, the development of the OG will be a never-ending process. Moreover, the Paris 2024 OG will have a great impact. If Paris 2024 is successful and the 2026 WOG succeed in realizing the OA's principle promises, then a new era in the sports world will begin. The future OG and thus a new OH—more intangible, emotional and oriented towards the needs of the host cities—will become reality. Wishful thinking?—Maybe but the attempt of the IOC is ambitious and the Olympic Movement is likely to manage the turnaround successfully; future research will need to analyse what will and what has become reality.

**Author Contributions:** Conceptualization, M.S.; Methodology, M.S.; Software, L.H.; Validation, M.S. and L.H.; Formal analysis, L.H.; Investigation, M.S. and L.H.; Resources, M.S.; Data curation, L.H.; Writing—original draft preparation, M.S.; Writing—review and editing, M.S.; Visualization, M.S.; Supervision, M.S.; Project administration, M.S.; Funding acquisition, M.S.

**Funding:** This research received no external funding.

**Acknowledgments:** We kindly thank the experts for their time and information given in the interviews.

**Conflicts of Interest:** The authors declare no conflict of interest.

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
