# Peer review of "Does the Olympic Agenda 2020 Have the Power to Create a New Olympic Heritage? An Analysis for the 2026 Winter Olympic Games Bid"

_sustainability, doi:10.3390/su11020442_

Round 1

Reviewer 1 Report

Referee Report – Sustainability – 408974

Does the Olympic Agenda 2020 Have the Power to Create a New Olympic Heritage? An Analysis for the 2026 Olympic Winter Games Bid

This paper surveys a number of stakeholders related to the Olympics to ask whether the IOC’s Agenda 2020 is likely to make a difference in improving the economic sustainability of the 2026 Winter Olympics.

The main results are that the 2026 Winter Olympics bidders are submitting bids with much lower budgets and are much more spread out than previous Games. The paper then summarizes the opinions of the interviewees.

The paper is well-written and interesting reading. I do have some concern about whether this is research or simply journalism. Interviewing a bunch of hand-selected people and reporting what they said is not precisely scientific inquiry, at least in economics, but it may have a place in a special interdisciplinary issue on culture and heritage management.

My recommendations is to somewhat increase the quantitative aspects of the paper. For example, at line 31, it would be possible to systematically but briefly run through the bidders that dropped out of the race to host both SOG 2024, WOG 2022, and maybe the WOG 2026, although that process is still ongoing. The authors do something like this around line 74, but unless you are an expert in the field, one would have no idea whether a handful of cities dropping out is normal and whether the IOC was left with a sufficient number of remaining bidders. Of course, one of the major issues in the 2022 WOG bidding is that all of the democracies, where the people had a say about whether they wanted the WOG to come, dropped out and only Beijing and Almaty, not exactly shining beacons of democracy, remained.

And again it would be nice to highlight in the introduction, around line 37 exactly how costly the Olympics have become. (Or maybe be a bit more explicit about the costs of some of the recent games such as Sochi, Rio, Beijing, and Athens around line 64. Maybe even mention Nagano where the final expenses will never be known since the organizers destroyed the records at the end of the event. This plays into the transparency argument.)

Line 244, did the authors previous define MICE. I missed it and had to look it up on the internet. And does Barcelona really beat out London and Paris for this sort of tourism? I find that surprising even though I am well-aware of Barcelona’s growth as a tourist destination from roughly 10th in Europe in the 80s to roughly 4th now.

Table 2 is a nice addition. For me it was the most interesting piece of the paper as one can clearly see the changes. Might I suggest two final lines with averages for 2002-2018 compared with the post Agenda 2026 bids. Not sure where to include the Beijing bid. It is pre-Agenda 2020, but so many of the other bidders dropped out, it is hard to say if it is representative of the old or new regime. Maybe just exclude.

All of the quotes from line 376-487. Interesting, but not really academic. I guess I can live with this material as it is the primary original contribution of the paper, but again it is more like journalism than academic research.

In section 5.3, a huge limitation of the paper is that it is essentially impossible in this sort of research to get a random representative sample of stakeholders. Therefore, any findings may suffer both from selection bias in the authors’ choice of interviewees, as well as reporting bias in terms of what the authors’ report. This is, of course, part of my concern about whether the paper qualifies as scientific academic research. I would advocate that the authors include the non-random sample or selection bias problem as a limitation of this section.

In section 5.4, don’t we already have some evidence that Agenda 2020 is working as the IOC accepted both the LA and Paris bids in order to reduce the costs of rebidding for 2028. And, to my knowledge both of those bids are fairly austere by modern standards. The two 2026 bids also appear somewhat scaled back in comparison to recent WOG as well, right? Still, I think what several of the interviewees are saying is we won’t really know whether the IOC is serious about Agenda 2020 until they are faced with a lavish, economically unsustainable bid against a cheaper but more responsible bid and they decide to go with the more reasonable bid.

Author Response

Please find attached the Response to Reviewer 1

Reviewer 2 Report

The topic is highly relevant and will suit the journell well. Especially because it is about a sustainability beyond the enviromental topic. The method meets the research questions well and you did a lot of interview work to achive your results. 

I have only Little concerns with the text that can easily be corrected: 

It is hard to read text with too much abbreviations like OC, RQ and so on. Please write less abreviations.  

110 Thomas Bach - the newly elected IOC President --> after fife years you cannot call him newly elected anymore. I would write something like "after beeing elected Thomas Bach first step was to start a new Olympic Agenda" 

Author Response

Please find attached the Response to Reviewer 2
